# Transformer-Based Distillation Hash Learning for Image Retrieval

Yuanhai Lv [1,2], Chongyan Wang [3], Wanteng Yuan [4], Xiaohao Qian [1], Wujun Yang [2,*] and Wanqing Zhao [1]

1    School of Information Science and Technology, Northwestern University, Xi'an 710127, China
2    Information Network Center, Xi'an University of Posts and Telecommunications, Xi'an 710121, China
3    Social Cooperation Department, Xi'an University of Finance and Economics, Xi'an 710100, China
4    Network Technology Department, Xi'an Aeronautics Computing Technique Research Institute, Xi'an 710065, China
*    Correspondence: wujun@xupt.edu.cn

**Abstract:** In recent years, Transformer has become a very popular architecture in deep learning and has also achieved the same state-of-the-art performance as convolutional neural networks on multiple image recognition baselines. Transformer can obtain global perceptual fields through a self-attention mechanism and can enhance the weights of unique discriminable features for image retrieval tasks to improve the retrieval quality. However, Transformer is computationally intensive and finds it difficult to satisfy real-time requirements when used for retrieval tasks. In this paper, we propose a Transformer-based image hash learning framework and compress the constructed framework to perform efficient image retrieval using knowledge distillation. By combining the self-attention mechanism of the Transformer model, the image hash code is enabled to be global and unique. At the same time, this advantage is instilled into the efficient lightweight model by knowledge distillation, thus reducing the computational complexity and having the advantage of an attention mechanism in the Transformer. The experimental results on the MIRFlickr-25K dataset and NUS-WIDE dataset show that our approach can effectively improve the accuracy and efficiency of image retrieval.

**Keywords:** image retrieval; Transformer; self-attention; knowledge distillation; hashing learning





## 1. Introduction

With the development of WEB 2.0, Internet information generation has changed from traditional website employee generation to user-led generation. Virtual social activities with the help of various multimedia social platforms are becoming increasingly normalized, and the human-oriented self-media have led to a qualitative change in the status and role of people in Internet activities. From the traditional information receiver to the information publisher, more and more people are communicating by publishing and sharing multimedia information. Birjandi et al. proposed the use of text to retrieve image content (KBIR, keyword-based image retrieval) to meet the needs of users for large-scale image retrieval [1]. By manual annotation, the method uses keyword attributes between text and images to build an index. However, manual annotation is labor-intensive in many cases. Users are often unable to describe the content of an image, so researchers have proposed a technique that allows users to enter images to search for relevant images (content-based image retrieval—CBIR) [2]. These methods have automatically extracted the image content features by analyzing the image content and then quantifying them, and building indexes for retrieval based on the quantified content features. However, because traditional content-based image retrieval methods usually use manual features, which is a fixed representation of visual features, and lack the ability to learn, retrieval performance is challenging to improve. With the study of deep convolutional neural networks and the large-scale accumulation of image data, many methods are using deep convolutional neural networks (DCNNs) to automatically learn the features of images and use these features to retrieve images from large-scale datasets. Peng et al. proposed an image retrieval method

based on DCNNs and binary hash learning [3]. The method uses DCNNs to learn the intrinsic distribution of images and extract image features while adding a hashing layer to the DCNNs to learn deep features and hash codes to perform an effective retrieval.

In recent years, the Transformer has become a prevalent architecture in the field of deep learning. For example, BERT [4] and GPT-2 [5], which have become very famous works in the natural language processing (NLP) field [6] in recent years, use the Transformer architecture. Transformer relies on a simple but potent mechanism, the Attention mechanism [7], which allows neural network models to selectively focus on certain parts of the input and reason more efficiently. Various architectures based on Transformer have been used with encouraging results in sequence prediction, language modeling, and machine translation.

Convolutional computation has better locality and spatial invariance and has an inherent natural advantage (inductive bias) for visual problems. The CNN model [8] needs to obtain a larger perceptual field by continuously stacking Conv layers [9]. Moreover, the number of operations required to compute the association between two locations increases with the location distance. The self-attention is the fundamental component of Transformer [7], and the number of operations required by self-attention to compute the association between two positions is distance independent. The inherent advantage is that, unlike convolution which has a fixed and limited field of perception, the self-attention operation can obtain long-range information [10]. By examining the weight distribution of each Attention Head, self-attention can produce more interpretable results.

Nowadays, inspired by the Transformer in NLP, researchers have extended the Transformer to the field of computer vision (CV). Dosovitskiy et al. proposed ViT [11], a model that feeds a sequence of image blocks (Patches) into a standard Transformer. This framework achieves the same state-of-the-art results on multiple computer vision task baselines. Some Transformer architectures for image retrieval tasks have also been gradually proposed [12]. These apply Transformer to pre-trained models for feature extraction, and then perform a similarity comparison in the feature space to solve problems in image retrieval.

Transformer can obtain the global sense field by self-attention mechanism, and can enhance the weight of unique discriminable features for image retrieval tasks to improve the retrieval quality. However, Transformer is computationally intensive, and it is challenging for it to meet real-time requirements when used for retrieval tasks. Knowledge distillation can replace a teacher model with a lightweight student model to improve speed and ensure accuracy. Therefore, it is necessary to design a framework based on knowledge distillation [13] to improve the retrieval speed for the Transformer-based image retrieval model. In recent years, the parameter size of models has become larger and larger, which often requires a large number of memory resources and is time-consuming to run during the deployment phase of the model. By model distillation, the model with huge parameters is compressed into a small parametric number model, which can make the model occupy fewer resources and be less time-consuming without losing accuracy.

This paper proposes a Transformer-based image hash learning framework, and the constructed framework is compressed for efficient image retrieval using knowledge distillation. First, Resnet-based Backbone is used to extract the image features. Then, the features are input to multi-head of Transformer as query, key, and value, respectively, and the mutual attention weights are calculated using Transformer. Then, the features fused by Transformer are mapped to Hamming space to perform more compact hash learning. Finally, the performance of this model is distilled into a smaller and faster student model for real-time retrieval.

The main contributions of the proposed method include: (1) by introducing the self-attention mechanism of the Transformer model, it makes the image features a global perspective, which can strengthen the weights of unique discernible features and improve the image retrieval quality. (2) By introducing knowledge distillation, the computationally heavy Transformer model is compressed into an efficient lightweight model, which reduces the computation and also has the advantage of features learned by the Transformer. (3) The

experimental results on two public datasets achieved encouraging results, demonstrating the effectiveness of the proposed method.

## 2. Related Works

In recent years, many machine learning-based search methods have been proposed to perform an efficient search of multimedia data. Among them, hash-based methods are gradually becoming the majority approaches for this problem [14–19]. These methods mainly use a variety of hash functions to encode the high-dimensional feature of an image to a low-dimensional representation, while expecting to preserve the approximate relationships in the original space after mapping them. Traditional hashing methods usually use manual image features such as SIFT [20], HOG [21], etc., to extract image features and then transform the features into hash codes using a fixed hash mapping function. This will reduce the capacity to express visual content and cannot handle complex similarity semantics well. Chugh et al. [22] combined multiple features to improve the retrieval of plants. With the popularity of deep convolutional neural networks (CNNs), CNNs are also gradually used for hash learning to solve the above problems. CNN-based supervised hash learning methods have achieved groundbreaking experimental results on many baselines [17–19,23–25]. For example, Xia et al. [23] proposed a CNN-based hashing method that learns binary hash codes by supervised training and demonstrates significant search performance on some public baselines. Zhang et al. [24] proposed to increase the rule element of the loss function based on triplet learning for the supervised deep hash coding method by using a Laplacian matrix. Furthermore, the method achieves bit scalability by giving a weight to each bit of the hash coding. The approach proposed by Zhao et al. [19] learns the hash coding of objects, while the method implements weakly supervised hash coding using multi-instance learning.

Google first proposed the Transformer model in 2017, which was first proposed by Vaswani et al. [26], as an only attention-based mechanism to implement machine translation tasks. Subsequently, Devlin et al. [4] pre-trained the Transformer by letting the model predict the masked words on untagged text data. This approach (i.e., BERT) was later considered a new paradigm for natural language representation models. Inspired by the Transformer in NLP, researchers have extended this mechanism to the field of computer vision (CV). In contrast to the previous CNN models [27], Chen et al. [28] pre-trained a sequence Transformer to predict masked pixels and was much more effective than CNN on image classification tasks. Dosovitskiy et al. proposed ViT [11], which applies the standard Transformer to image block (Patch) sequences for learning the embedded representation sequences of blocks. The output of the Transformer encoder is used as a representation and prediction of the image, which makes it equally state-of-the-art performance on multiple image recognition baselines. Transformer also achieved the desired results in high-level vision (HLV) tasks. High-level vision tasks are concerned with understanding and using the semantic content in images [29]. DETR [30] attempts to solve the image object detection problem using the Transformer, which treats the object detection task as an image-to-set prediction problem and simplifies detecting images. To address the limitations of the Transformer Attention module in processing image feature maps, researchers have further proposed deformable DETR for end-to-end object detection [31]. Since DETR still heavily relies on object box prediction during the training process, Wang et al. [32] proposed a dual-path Transformer for end-to-end panoramic segmentation, which effectively unifies semantic segmentation and instance segmentation.

Transformer has also gained some attention in the field of image retrieval. Liu et al. proposed the first large-scale text-to-image retrieval (VisualSparta) based on Transformer [33]. The proposed method can retrieve relevant images from a large and unlabeled set of images under a given text query. Facebook proposed a visual Transformer-based image retrieval model [12]. The model uses a visual Transformer to generate image descriptors and trains the model with a metric learning objective. The metric learning objective combines contrast loss with a differential entropy regularizer. In the Google Landmark Recognition 2021 Kag-

gle competition, Henkel et al. [34] proposed an effective end-to-end network architecture for large-scale landmark recognition and retrieval. The network architecture combines DLOG (orthogonal Local and Global) [35] and the hybrid Swin–Transformer model [36] and uses a predictive retrieval approach. For each query image, the method uses L2 to normalize the cosine similarity between image descriptors and then searches for the most similar image in the indexed image database.

Although many Transformer-based models have been proposed to solve computer vision tasks such as image retrieval, existing Transformer models are usually huge and computationally expensive, e.g., the base ViT model [11] requires 18 billion FLOPs to process the images. The knowledge distillation algorithm [13] is a technique that can compress a large network into a small efficient network and obtain comparable performance. The knowledge distillation algorithm usually uses a teacher network (large model) to teach a student network (small model), transferring the knowledge from the teacher network to the student network so that the performance of the student model is as close as possible to the performance of the teacher model. In 2015, Hinton et al. [13] first proposed the concept of knowledge distillation (KD) in neural networks, using the output logits of the teacher network or the integrated network (many teacher networks) and applying these logits to train fast small networks. Remero et al. [37] used not only the final output logits of the teacher network, but also its intermediate hidden layer parameter values (intermediate representations) to train the student network. Mirzadeh et al. [38] introduced multi-step knowledge distillation in teacher networks and student networks with teaching assistant networks. Knowledge distillation can be used for many networks (e.g., from large to small networks, from single networks to integrated networks, from CNNs to Transformer, etc.). More relevantly, Tian et al. [39] used distillation learning to fuse multi-modal transformers for a sketch-based image retrieval task. Few studies are currently related to knowledge distillation on Transformer-based image hashing learning networks. Our work focuses on using Transformer to obtain hash codes with a global view and high uniqueness and using knowledge distillation to obtain lightweight and efficient models that can be deployed in practice.

## 3. Method

This section describes our proposed framework, including a Transformer model for learning hash representations with a global view and high uniqueness and a lightweight and efficient model obtained by distillation that can be used for practical deployment. Section 3.1 introduces the architecture of the proposed framework in this work. Section 3.2 describes a CNN-based backbone for image feature extraction. Section 3.3 describes a Transformer teacher module for high-level semantic hash learning and a lightweight and efficient student model. Section 3.1 introduces the training phase including a ranking loss based on triplet samples and a distillation learning loss.

### 3.1. Model Overview

This work first extracts the visual features of the images using a CNN-based backbone. Then, the decoder of Transformer is used to perform the fusion of different image patch features by self-attention. Moreover, the final fused features are mapped to Hamming space to perform more compact hash learning. Finally, the knowledge learned by the Transformer module is distilled into a smaller and faster model.

In this paper, backbone utilizes a pre-trained ResNet model [27]. The backbone model is mainly used for feature extraction on images, so that this neural network model is generic and can be integrated with any advanced deep model. These methods can also be trained on unlabeled or weakly labeled data, thus further improving their performance.

The cross-attention described in this work utilizes the decoder in the Transformer structure to perform cross-attention. In this method, the query, key, and value in the cross-attention layer in the standard decoder are all from the image features output by the

backbone, and the process of computing attention by all three completes the cross-attention among visual features and achieves feature fusion.

In hash learning, the fused features are mapped into compact pseudo-binary codes that are used to improve the efficiency of image retrieval. The same image features from the backbone are input in the student module. However, instead of performing Transformer's attention computation, the linear layer is directly used to perform accelerated projection and constrain the output to be consistent with the Transformer. Figure 1 illustrates the overall architecture of our proposed framework.

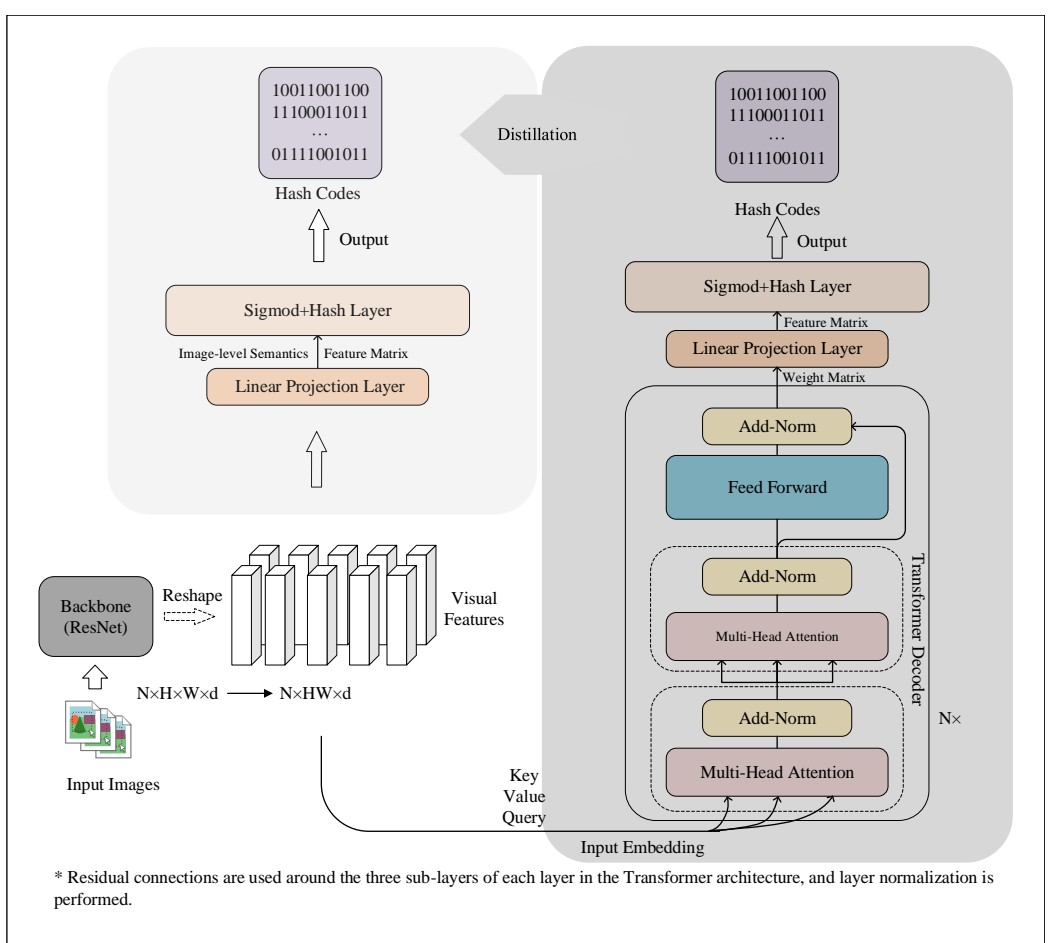

**Figure 1.** The overall architecture of our proposed framework.

### *3.2. CNN-Based Backbone*

In our work, the main role of the backbone is to perform the initial feature extraction of the image. Our backbone uses the ResNet-50 model [27]. Figure 2 is a brief framework diagram of ResNet-50. In ResNet-50, residual learning is applied to every few stacked layers to construct a residual block, defined as:

$$\mathbf{y} = F(\mathbf{x}, \mathbf{W}) + \mathbf{x}, \tag{1}$$

where $\mathbf{x}, \mathbf{y}$ are the input and output vectors for computing the current residual block. $F(\mathbf{x}, \mathbf{W})$ denotes the residual mapping function, and $\mathbf{W}$ is the parameters to be learned, $F(\mathbf{x}, \mathbf{W}) + \mathbf{x}$ is achieved by adding the shortcut linking layers and elements, and the dimensions must be equal before their summation.

In our work, we remove the final averaging pooling layer and full connection layer of Resnet-50, and use convolutional computation to extract its spatial features $F_0 = \mathbb{R}^{H \times W \times d}$, where $H \times W$ represent the height and weight of the feature map, respectively, and $d$ denotes the dimension of features. We set $H = W = 18$, so in total there are 324 elements

of feature embedding vectors. Since the output dimension of ResNet-50 is 2048, we set the dimension of our embedding size $d$ as 2048.

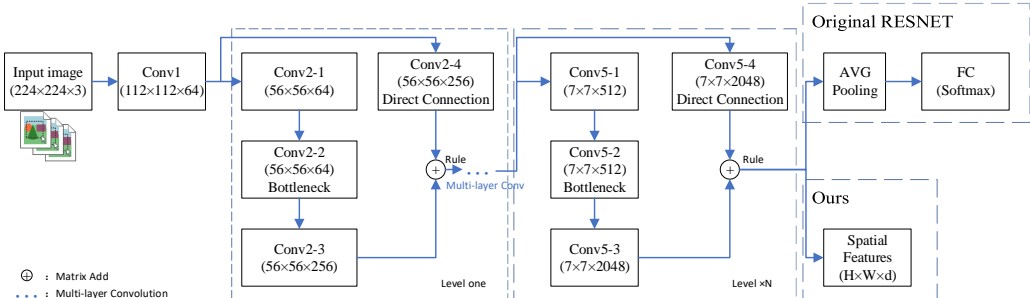

**Figure 2.** A brief framework diagram of ResNet-50.

### 3.3. Transformer Teacher Module and Student Module

**Query update:** The visual features $F_0$ of the image obtained from the above backbone are then input into the decoder of Transformer as key, value, and query. The image's spatial features are calculated using a multi-layer Transformer for self-attention. The input image is divided into $d$ embedding vectors, and each embedding vector obtains a one-dimensional position encoding token corresponding to its position and is used as input to the decoder regularly. The prior position is merged by adding a learnable one-dimensional position encoding code to the input embedding vectors. An extra learnable CLS token is added to the input embedding vector to represent its correspondence to the output token as a global concept. The Transformer consists of $L$ layers, each layer consisting of two main modules: a multi-headed self-attentive (MSA) layer, which applies the self-attentive operation to different embedding vectors of the inputs, and a feed-forward network (FFN). Both the MSA and FFN layers are preceded by a normalization layer, and followed by a skip connection layer. The query of the $(i)$-th decoder layer $Q_i$ is updated based on the output of its previous layer $Q_{i-1}$ as follows:

$$Self - Attention : Q_i' = Q_{i-1}' + MultiHead(\widetilde{Q}_{i-1}, \widetilde{Q}_{i-1}, Q_{i-1}) \tag{2}$$

$$FFN : Q_i = FFN(Q_i') \tag{3}$$

where $\widetilde{Q}_{i-1}$ denotes the embedding vectors with the position encoding of outputs at layer $(i-1)$-th, $Q_i'$ is the intermediate variable, $MultiHead(query, key, value)$ and $FNN(x)$ are the multi-headed attention mechanism and feed-forward network, respectively.

**Pooling:** We need to extract a compact code that globally describes the image. In our reference pooling approach, we directly treat the output of the CLS embedding as a global image descriptor.

**Reduction and binarization:** After obtaining the global image descriptors, we further perform dimensionality reduction and binarization to improve the retrieval speed. Specifically, we use a combination of a linear projection layer and a Sigmoid function to map the global image descriptors through the fully connected layer to a smaller size $b$ bit codes and project the logit values to $[0, 1]$ using the Sigmoid function:

$$\hat{\mathbf{h}} = Sigmoid(\mathbf{W}_k^T \mathbf{z}_{cls} + \mathbf{b}_k) \tag{4}$$

where $\mathbf{W}_k^T$ and $\mathbf{b}_k$ are the parameters of the linear projection layer and $\hat{\mathbf{h}}$ is the pseudo hash code. It is worth mentioning that since the symbolic function sgn() is not derivable, the pseudo hash codes generated in the training phase are real-valued, while in the inference phase, the binary hash codes are generated by the following formula:

$$\mathbf{h} = \frac{1}{2}(sgn(\hat{\mathbf{h}} - 0.5) + 1) \tag{5}$$

where **h** denotes the binary hash code.

**Student module:** The designed student model is straightforward. We directly remove the Transformer part of the teacher model. The feature map generated from the backbone will go through two fully-connected layers and output the same size as the teacher model.

*3.4. Training*

**Loss function for teacher model:** there are two losses used to train the teacher model, namely metric ranking loss based on triplet samples and quantified loss. The metric ranking loss function is used to make the similarity between positive sample pairs greater than the similarity between negative sample pairs:

$$L_{Triplet} = \sum_n [S(I_n, I_n^+) - S(I_n, I_n^-) + \delta]_+ \tag{6}$$

where $[x]_+ = max(0, x)$, $S()$ is the cosine similarity between sample pairs, $(I_n, I_n^+)$ are positive sample pairs and $(I_n, I_n^-)$ are negative sample pairs. The metric ranking loss would treat the sample pair of $S(I_n, I_n^-) + \delta > S(I_n, I_n^+)$ as a valid sample pair and increase the penalty, while for the sample pair of $S(I_n, I_n^-) + \delta \leq S(I_n, I_n^+)$ is considered to satisfy the desired goal and is therefore ignored. By training with this loss, the cosine identity of the negative sample pair will be guaranteed to be at least $\delta$ greater than the cosine identity of the positive sample pair.

The quantization loss in our approach is mainly to constrain the network output pseudo hash codes where the code values are as close to 0 or 1 as possible, as follows, which penalizes the network if the output of a neuron is close to 0.5:

$$L_{Quan} = -\sum_n \frac{1}{b} (\hat{\mathbf{h}}_n - 0.5\mathbf{1})^T \cdot (\hat{\mathbf{h}}_n - 0.5\mathbf{1}) \tag{7}$$

where **1** represents a vector of ones of length $b$. During training for the teacher model, we weight the two loss components $L_{Triplet}$ and $L_{Quan}$ by factors $\lambda_1$ and $\lambda_2$, respectively. Therefore, the overall loss for the teacher model is: $L_{Teacher} = \lambda_1 L_{Triplet} + \lambda_2 L_{Quan}$.

**Loss function for student model.** The student model uses the same backbone as the teacher model and both sides share the same backbone parameters. We fix the backbone parameters when training the student model. The subsequent structure is mapped to the same $b$ bits as the output of the teacher model by a fully connected mapping layer and activated with a sigmoid function. Relative entropy is used here as a distillation loss function to measure the distance between the two model distributions. Assuming that, for any sample $n$, the hash code output by the teacher model is $\hat{\mathbf{h}}_n^{tech}$ and the hash code output by the student model is $\hat{\mathbf{h}}_n^{stud}$, the distillation loss is

$$L_{distill} = \sum_n p_n \cdot (\log p_n - \log q_n) \tag{8}$$

$$\log p_n = \frac{\exp(((\hat{\mathbf{h}}_n^{tech})^T \cdot \hat{\mathbf{h}}_n^{tech})/\tau)}{\sum_n \exp(((\hat{\mathbf{h}}_n^{tech})^T \cdot \hat{\mathbf{h}}_n^{tech})/\tau)} \tag{9}$$

$$\log q_n = \frac{\exp(((\hat{\mathbf{h}}_n^{stud})^T \cdot \hat{\mathbf{h}}_n^{stud})/\tau)}{\sum_n \exp(((\hat{\mathbf{h}}_n^{stud})^T \cdot \hat{\mathbf{h}}_n^{stud})/\tau)} \tag{10}$$

where $p_n$ and $q_n$ denote the probability distributions of the corresponding sample $n$ in the teacher and student models, respectively, and $\tau$ is the temperature parameter in the knowledge distillation. After training with this loss, the student model will acquire a representation capability close to that of the teacher model, but with a more efficient inference speed.

## 4. Experiment and Analysis

### 4.1. Datasets and Metrics

**MIR Flickr-25K** [40]: In the MIR Flickr dataset, there are 25,000 images and 38 concepts of ground truth labels. In our experiments, we selected images with at least 1 of the 38 labels. Thus, a total of 16,000 images were used for training and 2000 images for testing.

**NUS-WIDE** [41]: It is a large-scale dataset that can be used to evaluate multiple multimedia tasks. Much of their image data come from contributions from social media sites. The dataset contains 269,648 images and 81 manual labels as ground truth that can be used for performance evaluation. It also contains 5018 tags annotated by amateur users. In our experiments, we only used the 21 most common labels and all images associated with these labels. Thus, we form a training set of 100,000 images and a test set of 2000 images.

In our experiments, the Hamming distances of the query images and the images in the training set are used for ranking. We consider a correct retrieval result when there is an identification label in the query image and the returned image. Evaluation metrics include mean accuracy (MAP), precision, and recall.

### 4.2. Training

Our backbone is pre-trained on ImageNet. We trained our model including a teacher module and a student module using mini-batch gradient descent with a learning rate of 0.001 and a learning rate of 0.0001 to fine-tune the backbone. We also used the momentum term with the rate of momentum equal to 0.9. The weighing factors for the losses $\lambda_1$ and $\lambda_2$ are all set to 1.0 for all the experiments, which were determined by cross-validation.

### 4.3. Analysis of the Lifting Effect of Transformer and Distillation

In this section, we provide an experimental study on using the Transformer model and fast student model for retrieval. To analyze the effectiveness of the Transformer model and fast student model, we report the experimental results of our method with different modules on MIRFlickr-25K and NUS-WIDE datasets. Our main findings are summarized below.

**Transformer teacher model is better than the student model in retrieval accuracy.** Table 1 compares the various variants of our approach. We observe that the attention-based Transformer model outperforms the linear projection-based fast student model. This suggests that the self-attentive mechanism in Transformer has the potential to improve the ability to discriminate features and thus improve retrieval quality.

**The student model has significant advantages over the teacher model in terms of retrieval efficiency.** As we can see from Table 1, although the Transformer-based teacher model achieves high accuracy, it is too slow to generate hash codes. On the other hand, the fast student model achieves nearly 10 times the computational speed of the teacher model. This is because Transformer is computationally intensive. There exists a large amount of computation in the self-attention process.

**Table 1.** Transformer-based teacher model and fast student model comparison. Length is the size of hash codes. Time indicates the time for the model to generate the hash codes on the query image.

| Model | Train Data | Length | MAP | Time |
|-------|-----------|--------|--------|--------|
| Teacher | MIRFlickr-25K | 24 | 0.7582 | 2.11 s |
| | | 48 | 0.7435 | 2.18 s |
| Student | | 24 | 0.7112 | 0.19 s |
| | | 48 | 0.7088 | 0.21 s |
| Teacher | NUS-WIDE | 24 | 0.6932 | 2.11 s |
| | | 48 | 0.6882 | 2.18 s |
| Student | | 24 | 0.6473 | 0.19 s |
| | | 48 | 0.6482 | 0.21 s |

**Distillation improves the student model.** In Table 2, we compare two methods on NUS-WIDE and MIRFlickr-25K datasets, one with no teacher guidance and directly using the loss functions Equations (6) and (7) to train the student model, and the other with teacher guidance for distillation training denoted as Transformer. As seen in Table 2, the distillation method improves the performance of the student model by more than 10% on MAP when trained on NUS-WIDE, which significantly reduces the gap between the slow and fast models. On the other hand, the improvement when trained on MIRFlickr-25K is not as significant as on NUS-WIDE, probably due to the more minor training data of MIRFlickr-25K. In addition, the teacher model has limited ability to generalize the knowledge learned on the small training data, and thus has a lower upper limit when distillation learning.

**Table 2.** Distillation experiment with the Transformer-based model as the teacher and the linear projection-based fast model as the student.

| Model | Teacher | Train Data | Length | MAP |
|---|---|---|---|---|
| Student | None | MIRFlickr-25K | 24 | 0.6678 |
| | Transformer | | 24 | 0.7112 |
| | None | | 48 | 0.6561 |
| | Transformer | | 48 | 0.7088 |
| Student | None | NUS-WIDE | 24 | 0.5321 |
| | Transformer | | 24 | 0.6473 |
| | None | | 48 | 0.5396 |
| | Transformer | | 48 | 0.6482 |

*4.4. Comparison to the State of the Art*

In this session, we compared our method with the state-of-the-art methods performed using the same evaluation metrics. The contrasting methods include LSH [42], ITQ [16], SH [15], PCAH [43], SpH [44], DH [45], DeepBit [46], DSH [47], BDNN [48], DVB [49] and DOH [50]. The LSH, ITQ, SH, PCAH, SpH, and DSH are not deep learning based hashing methods. We extracted depth features from the pre-trained ResNet model [27] and used them as input for these methods in order to make a fair comparison. For the deep learning-based methods, i.e., DH, DeepBit, BDNN, and DOH, we evaluated them with the hyperparameter settings suggested in their papers and ran the source codes provided by the authors. For DVB [49], we directly refer to the results of the original paper.

The MAP results calculated using different lengths of hash codes on two datasets, MIRFlickr-25K and NUS-WIDE, are reported in Table 3. The reported results show the superiority of the proposed method and validate that the motivation of the proposed method is valid. Furthermore, by incorporating the self-attention mechanism of the Transformer model, image hash codes can be made global and unique. At the same time, this advantage can be instilled into the efficient lightweight model by knowledge distillation, which reduces the computational effort and has the feature advantage of the Transformer.

**Table 3.** MAP results for the MIR Flickr-25K and NUS-WIDE datasets, using hash codes of different lengths, were calculated using the first 5000 images retrieved.

| Methods | MIR Flickr-25K | | | | NUS-WIDE | | | |
|---|---|---|---|---|---|---|---|---|
| Length | 12 | 24 | 32 | 48 | 12 | 24 | 32 | 48 |
| LSH [42] | 0.5763 | 0.6065 | 0.5966 | 0.6263 | 0.3523 | 0.4096 | 0.4186 | 0.4555 |
| SH [15] | 0.6621 | 0.6433 | 0.6296 | 0.6225 | 0.5652 | 0.5061 | 0.4866 | 0.4546 |
| SpH [44] | 0.5982 | 0.5832 | 0.5831 | 0.582 | 0.4656 | 0.4662 | 0.4473 | 0.4481 |
| ITQ [16] | 0.6932 | 0.7082 | 0.6686 | 0.6991 | 0.6332 | 0.6255 | 0.5922 | 0.6481 |

**Table 3.** *Cont.*

| Methods | MIR Flickr-25K | | | | NUS-WIDE | | | |
|---|---|---|---|---|---|---|---|---|
| PCAH [43] | 0.6444 | 0.6321 | 0.6377 | 0.6534 | 0.5775 | 0.5052 | 0.4921 | 0.4924 |
| DSH [47] | 0.6962 | 0.7076 | 0.6851 | 0.6612 | 0.5944 | 0.5987 | 0.5725 | 0.5795 |
| DH [45] | 0.6021 | 0.6176 | 0.6144 | 0.6174 | 0.4745 | 0.4631 | 0.4625 | 0.4755 |
| DeepBit [46] | 0.5887 | 0.6033 | 0.6092 | 0.6091 | 0.5465 | 0.5551 | 0.5626 | 0.5612 |
| BDNN [48] | 0.6654 | 0.6692 | 0.6678 | 0.6695 | 0.5932 | 0.5922 | 0.5912 | 0.6098 |
| DVB [49] | - | - | - | - | - | - | 0.562 | - |
| DOH [50] | - | - | 0.6728 | 0.6712 | - | - | 0.6145 | 0.6251 |
| Ours—student | 0.7046 | 0.7112 | 0.7092 | 0.7088 | 0.6361 | 0.6473 | 0.6526 | 0.6482 |
| Ours—teacher | 0.7471 | 0.7582 | 0.7485 | 0.7435 | 0.6818 | 0.6932 | 0.6925 | 0.6882 |

Figures 3 and 4 show the performance curves of retrieval results on the MIRFlickr-25K dataset and the NUS-WIDE dataset. It can be seen from these two figures that the proposed method outperforms all the compared methods on both datasets. The results express the superiority of our method over the compared methods.

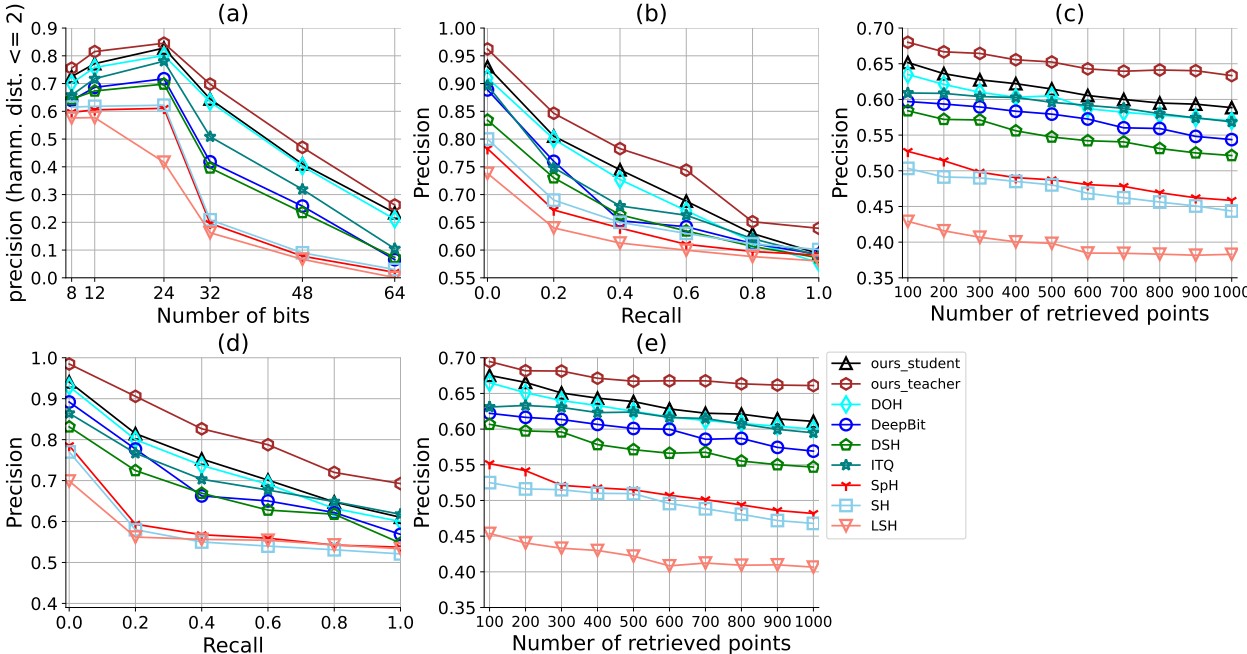

**Figure 3.** Performance curves of retrieval results on the MIRFlickr-25K dataset. (**a**) Precision using hash lookup within the Hamming radius 2; (**b**) Precision–recall curve for 48 bits; (**c**) Precision curve for 48 bits; (**d**) Precision–recall curve for 24 bits; and (**e**) Precision curve for 24 bits.

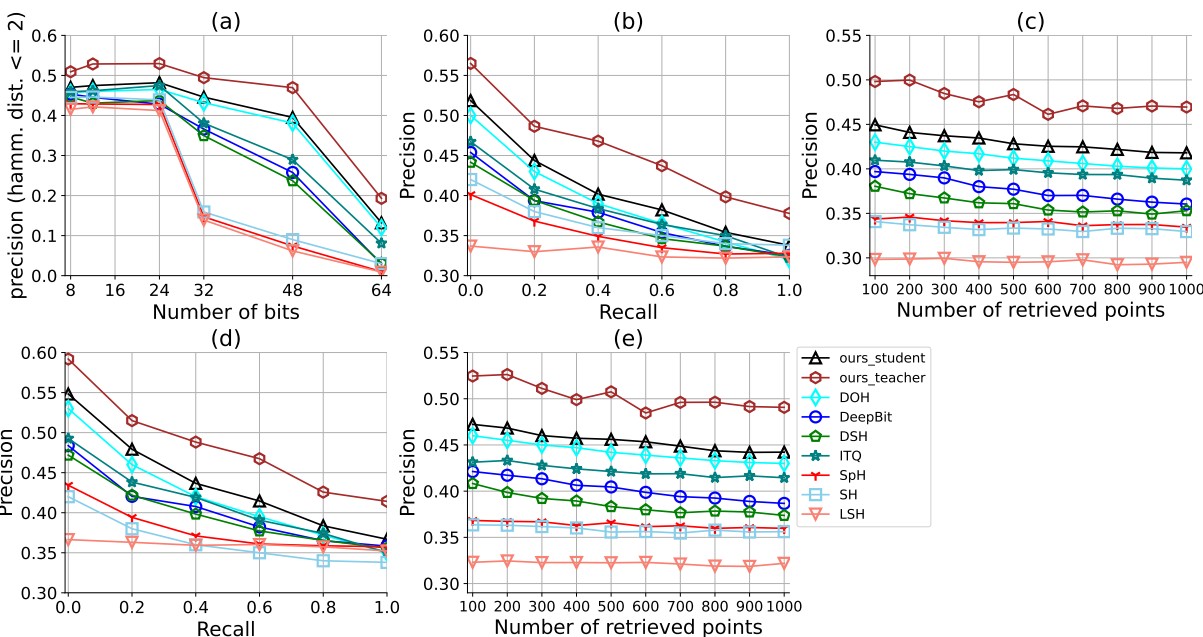

**Figure 4.** Performance curves of retrieval results on the NUS-WIDE dataset: (**a**) Precision using hash lookup within Hamming radius 2; (**b**) Precision–recall curve for 48 bits; (**c**) Precision curve for 48 bits; (**d**) Precision–recall curve for 24 bits; and (**e**) Precision curve for 24 bits.

## 5. Conclusions

This paper proposes a Transformer-based image hash learning with a knowledge distillation framework. By combining the self-attention mechanism of the Transformer model, the image hash code is enabled to be global and unique. At the same time, this advantage is instilled into the efficient lightweight model by knowledge distillation, thus reducing the computation and having the advantage of the Transformer's features. Experimental results on MIRFlickr-25K and NUS-WIDE datasets show that our approach can effectively improve the accuracy and efficiency of image retrieval.

**Author Contributions:** Funding acquisition, C.W. and W.Y. (Wujun Yang); Methodology, Y.L.; Resources, W.Y. (Wanteng Yuan); Writing—original draft, X.Q.; Writing—review & editing, W.Y. (Wujun Yang) and W.Z. All authors have read and agreed to the published version of the manuscript.

**Funding:** This research was supported by the China University Industry-University-Research Innovation Fund (Grant No. 2021FNA03001), the Research Project on Postgraduate Education and Teaching Reform of Xi'an University of Posts and Telecommunications (Grant No. YJGJ202034), the fellowship of China Postdoctoral Science Foundation (Grant No. 2020M683695XB), Shaanxi Provincial Philosophy and Social Science Research Project in Major Theoretical and Practical Issues (Grant No. 2022ND0181), and Xi'an Social Science Planning Fund Project (Grant No. 22LW44).

**Institutional Review Board Statement:** Not applicable.

**Informed Consent Statement:** Not applicable.

**Data Availability Statement:** Not applicable.

**Conflicts of Interest:** The authors declare no conflict of interest.

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
