# Peer review of "Transformer-Based Distillation Hash Learning for Image Retrieval"

_electronics, doi:10.3390/electronics11182810_

Round 1
Reviewer 1 Report
The paper presents a Transformer-based architecture for image retrieval. Two networks are trained mutually to allow a compressed and faster computation. This is technically sound, but the novelty seems not too big. The performance looks good, but the references for comparison are a bit out of date.
It is recommended to do the following changes before it is get accepted:
1. More recent related work (<5 years) should be used for comparisons.
2. In theory, a similar idea can be used to build a 'student network' which skips the transformer calculation from the 'teacher network'. Can you give us a discussion about it?
Reviewer 2 Report
Authors proposed a Transformer-based image hash learning framework and compress the constructed framework to perform efficient image retrieval using knowledge distillation.
Figures are of good quality but some points needs to be increased.
The overall presentation of the work needs to be enhanced.
How the proposed work is different from "TVT: Three-Way Vision Transformer through Multi-Modal Hypersphere Learning for Zero-Shot Sketch-Based Image Retrieval"
Related work needs to consider some good work like: a) An Image Retrieval Framework Design Analysis Using Saliency Structure and Color Difference Histogram b) A hybrid convolutional neural network model for diagnosis of COVID-19 using chest X-ray images
Round 2
Reviewer 2 Report
Acceptable